# Phenotypic Ultra-Rapid Antimicrobial Susceptibility Testing for Ceftazidime–Avibactam: In Support of Antimicrobial Stewardship

**DOI:** 10.3390/microorganisms13020414

**Published:** 2025-02-13

**Authors:** Inês Martins-Oliveira, Blanca Pérez-Viso, Rosário Gomes, David Abreu, Ana Silva-Dias, Rafael Cantón, Cidália Pina-Vaz

**Affiliations:** 1FASTinov, S.A., 4200-135 Porto, Portugal; ioliveira@fastinov.com (I.M.-O.); rosariogomes@fastinov.com (R.G.); dabreu@fastinov.com (D.A.); anadias@fastinov.com (A.S.-D.); 2Servicio de Microbiología, Hospital Universitario Ramón y Cajal, Instituto Ramón y Cajal de Investigación Sanitaria (IRYCIS), 28034 Madrid, Spain; blancapv45@gmail.com (B.P.-V.); rafael.canton@salud.madrid.org (R.C.); 3CIBER en Enfermedades Infecciosas, Instituto de Salud Carlos III, 28029 Madrid, Spain; 4RISE-Health Department of Pathology, Faculty of Medicine, University of Porto, 4200-319 Porto, Portugal; 5Division of Microbiology, Department of Pathology, Faculty of Medicine, University of Porto, 4099-002 Porto, Portugal

**Keywords:** antimicrobial susceptibility testing, ceftazidime–avibactam, Gram-negative bacteria, rapid diagnostics, flow cytometry, antibiotic escalation/de-escalation

## Abstract

Ceftazidime–avibactam (CZA) is a potent broad-spectrum drug combination covering extended-spectrum β-lactamases, AmpC, and carbapenemases of class A and D, OXA-48-type producers. Rapid antimicrobial susceptibility testing is crucial for the timely de-escalation/escalation of therapy. We evaluate CZA susceptibility using the CE-IVD FASTgramneg kit (FASTinov^®^), a ground-breaking 2 h assay, based on flow cytometry technology for antimicrobial susceptibility testing. The assay involved rapid bacterial extraction and purification from positive blood cultures (PBCs), followed by a 1 h 37 °C incubation and flow cytometry analysis (Cytoflex, Beckman-Coulter). The susceptibility report was generated using a proprietary software and interpreted using EUCAST and CLSI 2024 criteria. Sensitivity and specificity were calculated against a reference standardized method (disk diffusion) according to ISO20776-2:2021. Overall, 135 Enterobacterales and 73 *Pseudomonas aeruginosa* isolates were studied. Thirty-four isolates were resistant to CZA, including six *P. aeruginosa* and 28 Enterobacterales (24 metallo-beta-lactamase producers, three KPC variants, and one co-producing KPC+NDM). Sensitivity and specificity reached 100% when using EUCAST and CLSI criteria compared with the reference method. The FASTinov ultra-rapid susceptibility assay for CZA demonstrated excellent results, potentially enabling de-escalation/escalation even before the second dose. Combining the speed of a molecular assay with the comprehensive information of a phenotypic test offers valuable insights for treatment decisions.

## 1. Introduction

Antimicrobial resistance (AMR) ranks among the top ten global healthcare challenges, as recognized by the World Health Organization (WHO) [1]. Referred to as a “silent pandemic”, AMR rates have exceeded projections, posing significant threats to medical practices such as transplants, immunosuppressive therapies, and oncologic treatments, as 40% of empiric antibiotic therapy fails [2]. The development of new antibiotics is critical, but equally important is the urgent need for rapid diagnostic tests [3].

Conventional antimicrobial susceptibility tests (ASTs) provide essential information regarding bacterial susceptibility but often rely on growth-dependent methods, which can be slow. When a blood culture flags positive, it is sub-cultured for further AST determination and a report usually comes two days after, which could be late depending on the critical clinical situation of the patients. Rapid AST results are crucial to avoid or be able to quickly adjust empirical therapy, ideally before the second dose is administered. Several assays have been developed to accelerate the response time, although most of the new and sophisticated technologies still rely on methods to detect growth, which typically takes around 6–8 h. Ceftazidime–avibactam (CZA) is a broad-spectrum antibiotic combination, effective against Gram-negative bacilli including Enterobacterales and *Pseudomonas aeruginosa*, even those harboring resistance mechanisms such as extended-spectrum beta-lactamase (ESBL) or carbapenemases (e.g., OXA-48-like, KPC) [4]. However, CZA is ineffective against metallo-beta-lactamase (MBL)-producing isolates and has low activity against *Acinetobacter baumannii* and *Stenotrophomonas maltophilia* species. Recently, KPC-producing Enterobacterales variants depicting CZA resistance have been described, including in Portugal and Spain [5,6,7], causing molecular characterization to not be enough to guide therapy and increasing the importance of a phenotypic assay.

The first CZA-resistant strain was a *K. pneumoniae* identified in 2015, the same year that CZA was commercialized. This isolate was obtained in a patient who had not received the CZA treatment [8]. The characterized resistance mechanisms related to CZA resistance were likely due to efflux pumps and porin mutations [9]. Since then, new KPC-type variants conferring resistance to CZA have been described [8]. Nevertheless, although loss or mutations in porins might also contribute to resistance to CZA in KPC-producing strains [10], the most common mechanism remains mutations in the genes that encode KPC enzymes [8].

FASTinov has developed an ultra-rapid phenotypic susceptibility assay that can provide a bacterial susceptibility profile in less than two hours, directly from blood cultures, as compared to the traditional two-day timeframe [11]. In regions with high AMR rates, such as Italy and Greece, CZA is used empirically for safety reasons [12,13]. A rapid AST, such as the FASTgramneg kit from FASTinov, could facilitate timely de-escalation to narrower spectrum antibiotics, enhancing treatment safety and efficacy or even allowing a quick escalation if needed.

## 2. Materials and Methods

### 2.1. Strains

A total of 208 bacterial strains with a known susceptibility profile to CZA were selected, including 135 Enterobacterales (71 *Klebsiella pneumoniae,* including the ATCC 700603, BAA 1705, BAA 1706, and NCTC 13443 strains), 19 *Escherichia coli* (including the ATCC 25922, ATCC 35218, ATCC 8739, BAA 2452, and NCTC 13476 strains), 16 *Serratia marcescens* (including the ATCC 14756 strain), 12 *Enterobacter cloacae* (including the CCUG 59627 strain), 5 *Proteus mirabilis*, 4 *Klebsiella aerogenes*, 3 *Salmonella enteritidis*, 2 *Enterobacter asburiae,* 1 *Providencia rettgeri* (BAA 2525 strain), *Enterobacter hormaechei*, and *Kluyvera cryocrescens* respectively), and 73 *P. aeruginosa* (including ATCC 27853). All ATCC strains were used as controls. Overall, 189 isolates belong to the FASTinov bacterial collection, including ATCC and clinical strains, and 19 of them were kindly provided by Ramón y Cajal Hospital, of which three were KPC mutants.

### 2.2. FASTinov Assay for CZA Susceptibility Determination

The FASTgramneg kit instructions were followed for CZA susceptibility determination, with samples taken directly from positive blood cultures (PBCs). A quick preparation of the sample in order to obtain a clean liquid colony was performed; briefly, 1 mL of the PBC was extracted and 50 µL of a haemolytic agent included on the kit was added and vortexed for 10 s. A centrifugation of 1 min at 13,000 rpm was performed and the supernatant was rejected. The pellet was hydrated with 500 µL of saline solution and put on the top of 500 µL of Histopaque^®^ (Sigma-Aldrich, St. Louis, MO, USA), a gradient separation reagent. The centrifugation was repeated and supernatant was also rejected. From this pellet, isolates were identified using MALDI-TOF (Bruker Daltonics, Bremen, Germany) according Cruz et al.’s protocol [14]. A bacterial suspension at 0.5 on the McFarland scale was prepared and then 1 mL was diluted in 7 mL of Muller–Hinton cation-adjusted broth medium (ref. 26177). One hundred µL of this cell suspension per well was used to hydrate a 96-well microplate panel previously prepared with 8 mg/L of CZA (breakpoint concentration) and a membrane depolarization fluorescent dye. Panels were incubated for 1 h at 37 °C with shaking.

### 2.3. Flow Cytometry Analysis

The flow cytometric analysis was conducted using Cytoflex (Beckman-Coulter, Brea, CA, USA), through appropriate templates. At least 30,000 cells were acquired and data were analyzed with a dedicated software, bioFASTv 2.1 (FASTinov proprietary), to determine the susceptibility phenotype (susceptible [S] or resistant [R] for Enterobacterales and *Pseudomonas aeruginosa*) according to EUCAST and CLSI breakpoints [15,16]. Several parameters were taken into consideration such as the size, complexity, and intensity of fluorescence of the treated cells to detect bacterial cell lesions due to antibiotic action through flow cytometry. Cells treated with CZA were compared with the viable control.

### 2.4. Internal Control of FASTinov Assay

The flow cytometric assay includes two controls: (1) a bacterial viable control (similar to a positive control in any AST assay) that consists of a well without CZA but with a fluorescent probe and (2) a dead control, in order to be sure that the fluorescent probe is working, with a well with a drug that kills bacteria and a fluorescent probe. Only assays where control 1 showing viable bacteria (low intensity of fluorescence) and control 2 showing dead bacteria (high intensity of fluorescence) were validated.

### 2.5. Reference Method for CZA AST

CZA antimicrobial susceptibility testing was determined with disk diffusion as a reference method. Disk potencies were 10/4 and 30/20 µg according to 2024 EUCAST and CLSI protocols, respectively [15,16].

### 2.6. Characterization of Resistant CZA Isolates

Resistant strains to CZA were phenotypically characterized, specifically regarding carbapenems. Minimal inhibitory concentration (MIC) to meropenem was determined using broth microdilution according ISO 20776-2:2021 [17] and results were interpreted according to EUCAST 2024 breakpoints. Additionally, the molecular detection of the main carbapenemase encoding genes (*bla*_VIM_, *bla*_KPC_, *bla*_OXA-48_ and *bla*_NDM_) was performed by PCR and further nucleotide Sanger sequencing, using primers and conditions previously described [18]. The analysis of the sequences was performed with Chromas 3.6.4 (Technelysium, Brisbane, Australia) and Vector NTI 10.3.0 (Invitrogen Corporation, Carlsbad, CA, USA). The BLAST (https://blast.ncbi.nlm.nih.gov/Blast.cgi; accessed on 10 February 2025) and Clustal Omega (https://www.ebi.ac.uk/jdispatcher/msa/clustalo; accessed on 10 February 2025) programs were used for comparison alignment between different sequences, respectively.

### 2.7. Statistical Analysis

The sensitivity and specificity of flow cytometry assay for CZA were determined against the reference method according to ISO 20776-2:2021.

## 3. Results

After sample prep from positive blood cultures, all the bacteria were correctly identified in at least one spot by MALDI-TOF. Out of the 208 strains, 34 were resistant to CZA (28 Enterobacterales and 6 *P. aeruginosa*) regarding the reference method and by flow cytometry; most resistant strains among Enterobacterales were MBL producers, these including 20 VIM, 3 NDM, and 1 IMP, respectively (Table 1). Furthermore, three CZA-resistant isolates were KPC producers and one of them was a co-producer of KPC+NDM (Table 1). Thirteen KPC-producing isolates were susceptible to CZA, these being the only effective drug against these multi-drug-resistant strains. We included in the study three *K. pneumoniae* isolates that were CZA resistant due to the presence of mutations on the *bla*_KPC-3_ gene (7), despite being susceptible to meropenem (see Table 1). Figure 1 illustrates a typical example of flow cytometry results of a CZA-susceptible and -resistant isolate, respectively. Flow cytometry histograms showed a significant rightward shift on the fluorescence intensity in the susceptible isolate when compared with the control; this shift indicates increased fluorescence intensity due to a cell lesion. In contrast, resistant strains showed no significant increase in fluorescence intensity when compared with the control. The sensitivity and specificity were 100% when using both EUCAST and CLSI interpretive criteria and compared with the reference method selected for this study.

## 4. Discussion

The combination of ceftazidime and avibactam was approved in the United States in 2015 and in Europe in 2016. CZA shows strong in vitro activity against KPC-producing Enterobacterales and has been linked to a reduced mortality rate in patients treated with it and in infections due to carbapenemase producers [19,20,21]. Despite its broad activity, concerns have been raised regarding the emergence of KPC-resistant mutants following CZA treatment [22,23], highlighting the significant capacity of this enzyme to evolve [8]. Phenotypic assays are the standard reference techniques for antimicrobial susceptibility determination. However, these methods are often slow and provide limited information and, traditionally, phenotypic tests require 48–72 h to deliver results. Molecular tests offer a solution to the time delay associated with phenotypic methods, as they can provide results within 1–5 h; nevertheless, these molecular methods are limited as they can only detect known genes and do not provide comprehensive information about the overall antimicrobial susceptibility pattern of the microorganism [24,25]. A phenotypic antimicrobial susceptibility assay is essential for the accurate selection of antimicrobial therapy. Most automated systems used in clinical microbiology laboratories still require up to two days to provide results from a positive blood culture, often delaying critical treatment decisions. The FASTinov kit offers a truly ultra-rapid solution, allowing therapeutic adjustments within the same shift—ideally before administering the second antibiotic dose [26,27].

On the other hand, CZA is a safe empirical treatment option when Gram-negative bacteria are isolated from blood cultures, but rapid phenotypic characterization of bacterial susceptibility to key antibiotics is crucial to target an appropriate therapy according to the AST results. In cases of CZA resistance, a rapid shift to alternative therapies can be made. If the bacteria are susceptible (as is the case in most instances), rapid AST enables swift de-escalation to narrower-spectrum antibiotics, which is vital to avoid prolonged CZA use (one of the primary reasons for discontinuing CZA). Without testing, relying solely on CZA treatment is no longer 100% reliable, as resistance has been reported even in KPC-producing strains [8] and extending the use of CZA beyond the necessary period could increase the risk of resistance development.

Apart from FASTinov, there are other marketed available tests, but normally the time to result is higher (5–8 h) when compared to those developed by FASTinov, as they rely on growth dependent methods [26].

Flow cytometry is a flexible and crucial tool in contemporary science and healthcare, offering quick and comprehensive cell analysis. This technology is typically used in immunology or hematology fields, and more recently has been in clinical microbiology laboratories. However, although a limitation could be access to flow cytometry equipment, this technology could represent an improvement in daily operations in the laboratory in response to bacterial infections.

FASTinov technology is disruptive as it is a phenotypic but not growth-dependent assay, which use flow cytometry to analyze multiple cellular parameters rapidly. To assess cell damage caused by antibiotic exposure using a fluorescent probe, a fully automated flow cytometric analysis was conducted with the CytoFLEX model B3-R0-V3 (Beckman Coulter, USA). The instrument is equipped with a blue laser (488 nm) and three fluorescent channels: 525/40 BP, 585/42 BP, and 690/50 BP. Additionally, this cytometer includes a plate reader for automatic analysis. A lot of cells are analyzed in a few seconds, with the collected data being significant. This approach, combined with a multiparametric software analysis, ensures accurate and automatic reporting, even directly from PBCs, avoiding a sub-culture step. A clean bacterial suspension was quickly prepared from the blood culture to be analyzed by flow cytometry, allowing for excellent identification under mass spectrometry. This information could be immediately provided to the clinician. A rapid AST needs rapid identification in order to allow a correct phenotypic classification. A significant population such as 30,000 cells is analyzed in 10 s. A dedicated software then compares several cell characteristics of the cells after drug exposure and compares these with non-treated cells. Excellent sensitivity and specificity results were obtained for CZA susceptibility determination both in Enterobacterales and *P. aeruginosa*, irrespective of bacterial species or different carbapenemase production, such as with MBL or KPC, and even those expressing mutated KPC carbapenemases displaying CZA resistance [8].

The qualitative assay described here is a breakpoint test, which could be considered a limitation of the assay, although FASTinov assays have demonstrated great potential to deliver quantitative information regarding MIC values, for instance to colistin [28] or vancomycin, if several serial dilutions of the drug are analyzed [11]. The qualitative categorization of susceptible/resistant information is usually enough for most clinical needs, allowing one to provide an ultra-rapid response to critical situations. A multicentric evaluation study showed excellent results with both spiked and patients’ blood cultures but the majority of isolated bacteria were susceptible [11]. Molecular characterization of carbapenemases has been used as a solution for antimicrobial treatment as information about the presence of a metallo-carbapenemase means resistance to CZA and in most occasions KPC producers means susceptibility. However, in the case of KPCs, certain mutations confer resistance to CZA [7].

Resistance to CZA remains limited, although mutations in KPC-2 and KPC-3 have been reported as showing resistance to CZA [8], but often remaining susceptible to carbapenems [29]. These mutations involve amino acid substitutions in beta-lactamases and changes in gene expression, often coupled with reduced outer-membrane protein expression or efflux pump overexpression [30].

The emergence of new KPC variants poses a significant challenge for clinical treatment, showing that molecular characterization is not enough or safe for patient treatment. While molecular tests are rapid, they can only detect known resistance mechanisms and do not report susceptibility profiles [3], so they should be combined with rapid phenotypic information.

## 5. Conclusions

The new rapid commercial flow cytometry AST performed directly from blood cultures showed high accuracy for CZA susceptibility determination. As a phenotypic assay, it represents an informative and safe tool for the management of critical patients in clinical settings, with a high potential clinical, economic, and social impact.

## Figures and Tables

**Figure 1 microorganisms-13-00414-f001:**
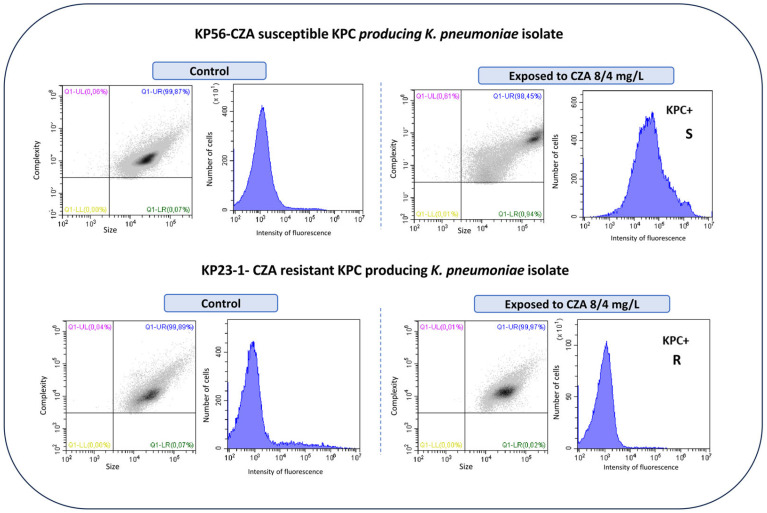
Scatter plots showing population distribution by size and complexity and flow cytometry histograms representing intensity of fluorescence of two KPC-producing *K. pneumoniae* isolates—one susceptible (top of image) and one resistant (mutant) to ceftazidime–avibactam (CZA); control population and cells exposed to 8 mg/L of ceftazidime associated with 4 mg/L of avibactam for 1 h. S-susceptible; R-Resistant.

**Table 1 microorganisms-13-00414-t001:** Carbapenemase characterization of ceftazidime–avibactam-resistant Enterobacterales isolates regarding disk diffusion, flow cytometry phenotypes, and molecular detection methods; isolate meropenem minimum inhibitory concentration (MIC) interval values.

Strain Details	Number of Strains	Disk Diffusion Phenotype	FC Phenotype	Carbapenemase Gene	Meropenem
					MIC (mg/L)
*K. pneumoniae*	2	R	R	*bla* _NDM-1_ *bla* _NDM-1_	>32
*E. coli*	1	R	R		2
*E. coli*	1	R	R	*bla* _IMP_	1
*E. cloacae*	2	R	R	*bla* _VIM-1_	16–64
*E. asburiae*	2	R	R	*bla* _VIM-1_	4–8
*E. asburiae*	1	R	R	*bla* _VIM-2_	2
*E. hormaechei*	2	R	R	*bla* _VIM-1_	8–>64
*K. pneumoniae*	1	R	R	*bla* _VIM-34_	64
*S. marcescens*	12	R	R	*bla* _VIM-1_	8–>64
*K. pneumoniae*	1	R	R	*bla* _KPC-62_	4
*K. pneumoniae*	2	R	R	*bla* _KPC-31_	0.12–0.25
*K. cryoscescens*	1	R	R	*bla* _NDM-1+KPC-2_	8

FC—flow cytometry; R—resistant.

## Data Availability

Data are contained within this article.

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
