# Peer review of "Phenotypic Ultra-Rapid Antimicrobial Susceptibility Testing for Ceftazidime–Avibactam: In Support of Antimicrobial Stewardship"

_microorganisms, 2025, doi:10.3390/microorganisms13020414_

Round 1
Reviewer 1 Report
Comments and Suggestions for Authors
The manuscript adequately describes an ultra-rapid susceptibility testing for the combination ceftazidime-avibactam (CZA). However, in some sentences it seems that CZA is one drug, when it is actually a combination of compounds (lines 16 and 51) (it should be corrected). On the other hand, references should be carefully revised to properly follow the journal's standards.
Author Response
Dear Editor and Reviewers of Microorganisms,
Many thanks for all your comments and suggestions, which have been used to improve our manuscript entitled ""Phenotypic Ultra-Rapid Antimicrobial Susceptibility Testing for Ceftazidime-Avibactam: A Support for Antimicrobial Stewardship” (Manuscript ID: microorganisms-3429003). This new version has been updated following the recommendations of the reviewers. Please, find below the point by point answers to the reviewer comments. All modifications from previous version are highlighted in yellow in the manuscript. All authors agree with the modifications introduced in the manuscript.
Reviewer comments:
Reviewer #1
Reviewer#1
1- The manuscript adequately describes an ultra-rapid susceptibility testing for the combination ceftazidime-avibactam (CZA). However, in some sentences it seems that CZA is one drug, when it is actually a combination of compounds (lines 16 and 51) (it should be corrected). On the other hand, references should be carefully revised to properly follow the journal's standards.
Authors: Thank you for your comment. You are right we have already changed that information.

Reviewer 2 Report
Comments and Suggestions for Authors
The MS "Phenotypic Ultra-Rapid Antimicrobial Susceptibility Testing for Ceftazidime-Avibactam: A Support for Antimicrobial Stewardship"
The assay involved rapid bacterial extraction and purification from positive blood cultures and flow cytometry analysis. Sensitivity and specificity reached 100% as compared to EUCAST.
There some main points
1- Is there any other rapid Antibiotic susceptibility tests available in the market for comparison?
2- how the authors calculated the suitable sample size as those types of tests require larger samples size
3- the sensitivity of the isolates performed by the disc traditional, standard methods was performed by no data analysis for comparison, or data interpretation and statistical analysis
4-PCR of resistant genes was not mentioned in the methods
5- PCR data the results do not involve all the isolates
6- How different variants of the genes in the table 1 was detected
7- The principle of the test needs to be more explained in the discussion section
8comparisons to disc and PCR with data analysis and interpretation need to be more explained with ref to previous studies.
Author Response
Dear Editor and Reviewers of Microorganisms,
Many thanks for all your comments and suggestions, which have been used to improve our manuscript entitled ""Phenotypic Ultra-Rapid Antimicrobial Susceptibility Testing for Ceftazidime-Avibactam: A Support for Antimicrobial Stewardship” (Manuscript ID: microorganisms-3429003). This new version has been updated following the recommendations of the reviewers. Please, find below the point by point answers to the reviewer comments. All modifications from previous version are highlighted in yellow in the manuscript. All authors agree with the modifications introduced in the manuscript.
Reviewer comments:
Reviewer #2
The MS "Phenotypic Ultra-Rapid Antimicrobial Susceptibility Testing for Ceftazidime-Avibactam: A Support for Antimicrobial Stewardship".
The assay involved rapid bacterial extraction and purification from positive blood cultures and flow cytometry analysis. Sensitivity and specificity reached 100% as compared to EUCAST.
There some main points:
1- Is there any other rapid Antibiotic susceptibility tests available in the market for comparison?
Authors: Thank you for your comment. Yes, these tests are available on the market, but they are not as fast as those from FASTinov. They take approximately 5-8 hours to deliver results, as they are growth dependent. For example, the dRAST test by Quantamatrix and the QuickMic test by Gradientech provide susceptibility testing, including results for ceftazidime/avibactam (CZA). We have included this information in the new version of the manuscript.
2- How the authors calculated the suitable sample size, as those types of tests require larger samples size
Authors: Thank you for your comment. We usually follow the ISO 20776-2:2021 for antimicrobial susceptibility validation. A lot of data has been generated already and resistant strains are not so frequent. This paper wants to highlight the benefits of phenotypic tests versus genotypic ones, including in our evaluation some KPC mutant strains that are resistant to CZA. We are always accumulating data from other validation sites.
3- The sensitivity of the isolates performed by the disc traditional, standard methods was performed by no data analysis for comparison, or data interpretation and statistical analysis
Authors: The AST assay described in this paper, is qualitative (S or R). For each isolate, results obtained were compared with the ones obtained by disk diffusion and sensitivity and specificity was calculated according to ISO 20776-2:2021. This is mentioned on Material and Methods section.
4-PCR of resistant genes was not mentioned in the methods
Authors: We apologize for this. Material and methods 2.6 section has been updated with the information about specific carbapenemase molecular characterization and a new reference regarding primers and conditions used has been also added.
5- PCR data the results do not involve all the isolates
Authors: Carbapenemase molecular characterization was performed in all CZA resistant isolates as described in Material and Methods section.
6- How different variants of the genes in the table 1 was detected
Authors: Thank you for your comment. After performing PCR and DNA electrophoresis and purification, the purified PCR product was subsequent sequenced by Sanger.
7- The principle of the test needs to be more explained in the discussion section
Authors: You are right we have already added that information.
8- Comparisons to disc and PCR with data analysis and interpretation need to be more explained with ref to previous studies
Authors: Thank you for your comment. We have now included it in the Discussion section of the manuscript.

Reviewer 3 Report
Comments and Suggestions for Authors
This is a study on the evaluation of phenotypic Ultra-Rapid Antimicrobial Susceptibility Testing for ceftazidime-avibactam compared with the reference method (disc diffusion test). Overall, the described method is innovative with the use of flow cytometry. I also agree that this method is rapid, and the technicians can obtain the result directly from the blood culture broth. I agree with the authors that direct PCR or antigen detection of blood culture broth has limitations, as it does not directly reflect the antibiotic susceptibility. Here are some of my comments:
1. The limitations of the current method should be mentioned. Flow cytometry is not readily available in all microbiology laboratories, therefore this test method may not apply to all laboratories, especially in resource-limited settings. Furthermore, polymicrobial bacteremia could affect the interpretation of the flow cytometry results, and this should probably be mentioned and elaborated on in the paper.
2. Lines 20-21 "based on detecting bacterial lesions through flow cytometry" should be re-written.
3. Occasional type-setting problems observed in this paper, including missing or extra spaces.
Author Response
Dear Editor and Reviewers of Microorganisms,
Many thanks for all your comments and suggestions, which have been used to improve our manuscript entitled ""Phenotypic Ultra-Rapid Antimicrobial Susceptibility Testing for Ceftazidime-Avibactam: A Support for Antimicrobial Stewardship” (Manuscript ID: microorganisms-3429003). This new version has been updated following the recommendations of the reviewers. Please, find below the point by point answers to the reviewer comments. All modifications from previous version are highlighted in yellow in the manuscript. All authors agree with the modifications introduced in the manuscript.
Reviewer #3
This is a study on the evaluation of phenotypic Ultra-Rapid Antimicrobial Susceptibility Testing for ceftazidime-avibactam compared with the reference method (disc diffusion test). Overall, the described method is innovative with the use of flow cytometry. I also agree that this method is rapid, and the technicians can obtain the result directly from the blood culture broth. I agree with the authors that direct PCR or antigen detection of blood culture broth has limitations, as it does not directly reflect the antibiotic susceptibility. Here are some of my comments:
- The limitations of the current method should be mentioned. Flow cytometry is not readily available in all microbiology laboratories, therefore this test method may not apply to all laboratories, especially in resource-limited settings. Furthermore, polymicrobial bacteremia could affect the interpretation of the flow cytometry results, and this should probably be mentioned and elaborated on in the paper.
Authors: Thank you for your comment. You are right, we failed to highlight the limitation of the test. We have now included it in the Discussion section of the manuscript.
- Lines 20-21 "based on detecting bacterial lesions through flow cytometry" should be re-written.
Authors: This sentence was re-written for a better understanding.
- Occasional type-setting problems observed in this paper, including missing or extra spaces.
Authors: Thank you for your comment. The entire manuscript has been reviewed to correct this typographic mistakes.

Round 2
Reviewer 2 Report
Comments and Suggestions for Authors
Dear editor and authors
The authors addressed some points however, there is still some important points and data with data analysis need to be added including
-Comparisons of the data obtained by this methods to data of disc and PCR with data analysis was not added.
-Sequencing data and gene bank numbers need to be added
Author Response
Dear Editor and Reviewer of Microorganisms,
Many thanks for your comment and suggestion, which have been used to improve our manuscript entitled ""Phenotypic Ultra-Rapid Antimicrobial Susceptibility Testing for Ceftazidime-Avibactam: A Support for Antimicrobial Stewardship” (Manuscript ID: microorganisms-3429003). This new version has been updated following the recommendation and we agree to convert to communication. Please, find below the point-by-point answers to the reviewer comments. All modifications from previous version are highlighted in yellow in the manuscript. All authors agree with the modifications introduced in the manuscript.
Reviewer#2 comments:
The authors addressed some points however, there is still some important points and data with data analysis need to be added including
1-Comparisons of the data obtained by this methods to data of disc and PCR with data analysis was not added
Authors: Thank you for your comment. We have already updated Table 1 of the manuscript for a clearer comparison between results obtained with each method
2-Sequencing data and gene bank numbers need to be added
Authors: Thank you for your comment. We have performed PCR and Sanger sequencing (alignment) in studied isolates, not WGS, so that the genomes have not been deposited in GenBank. We have explained the procedure regarding molecular characterization in a specific way to a better understanding for the reader.
